# Teachers and Learners' Perceptions of E-Learning Implementation in Special Times: Evaluating Relevance and Internationalization Prospects at Saudi Universities

**Mohammad Shariq** [1,*] , **Kh. Lutfy** [2,3] , **Ameen Alahdal** [4] **and Fahad Ibraheem Abdullah Aldhali** [5]

1 Department of English and Translation, College of Sciences and Arts at Methnab, Qassim University, Buridah 52571, Saudi Arabia
2 Department of Mathematics, College of Sciences, Taibah University, Medina 42353, Saudi Arabia; kazab@taibahu.edu.sa
3 Department of Mathematics, Faculty of Science, Zagazig University, Zagazig 44519, Egypt
4 Department of English and Translation, College of Science and Arts at Uglat Asugour, Qassim University, Buridah 52571, Saudi Arabia; a.alahdal@qu.edu.sa
5 Department of Quranic Studies, College of Sharia and Islamic Studies, Qassim University, Buridah 52571, Saudi Arabia; fthaala@qu.edu.sa
* Correspondence: m.aslam@qu.edu.sa

**Abstract:** The recent abrupt shift to a total e-learning modality has been a fresh yet daunting experience for educational institutions due to the onslaught of the COVID-19 pandemic. This shift has also raised the questions of relevance of educational modalities given the special times we are living in, as well as the happy possibility of universities gearing up for internationalization to prepare students for online learning. Before implementing these changes, however, in-depth study of the opinions and experiences of teachers and students at Saudi universities, among other parameters, is imperative. With this focus, the current study employs a mixed-methods research design from two universities in Saudi Arabia, namely Qassim University (QU) and Imam Abdulrahman Bin Faisal University (IAU). A purposive sample of 22 teachers and 54 students were the respondents, who were administered a questionnaire and interviewed at a later stage. Results showed that both teachers and students find the online teaching–learning experience challenging due to teacher-related, student-related and technology-related factors. The teachers rate themselves as being moderately competent in the use of technology tools for online learning, while students assess themselves as competent. Initiatives are offered by both students and teachers to improve the transition of universities to online education as well as to promote the quality of universities towards internationalization, particularly with living in the midst of a health crisis. The study has implications for curriculum implementers and designers committed to educational revolution. The outcomes of this proposed research can be the basis for relevance and internationalization initiatives of the selected universities in Saudi Arabia.

**Keywords:** e-learning; internationalization; perceptions; live-experiences; learning preferences

## 1. Introduction

Online and e-learning programs are becoming the biggest problem for many universities during the COVID-19 pandemic era. The pandemic triggered a digital revolution and curricular change (Naim & Alahmari, 2020) [1], as many institutions of higher education developed e-learning as the only way to continue their education in practice (Al-Ahdal, 2020; Muhammad et al., 2020) [2,3]. E-learning platforms can assist learning providers with learning processes management, planning, implementation and tracking. This investigation also seeks to assist educators, schools and colleges in encouraging learning during non-campus hours. In the wake of this pandemic, many of these devices came free of charge. Nevertheless, this abrupt transition to e-learning raised difficulties for university teachers and students. Students were not able to take to e-classes readily (Algahtani et al., 2020) [4]. As a

result, many students, teachers and parents questioned whether the transition to e-learning would be governed by the pandemic updates and how university education would be affected (Al-Hattami, 2020) [5]. Ever since the onslaught of the COVID-19 pandemic in Saudi Arabia, all universities moved to online courses. However, challenges have been many and diverse, which could have an adverse impact on learners' and teachers' experiences of the online teaching and learning delivery mode. These challenges include internet failure and the lack of expertise or information with respect to the use of other e-tools by teachers (Almaiah et al., 2020; Hoq, 2020) [6,7] and the difficulty of maintaining a high degree of participation and interest [8] (Balhareth et al., 2020). In addition, while online education tends to promote student autonomy, the engagement of Saudi students in online classes is often affected by previous experience with the traditional approach to teaching and training in Saudi Arabia, in which teachers played a central role and held complete authority over learners (Hamdan, 2014 [9]), which effectively restricted opportunities for active online student participation (Lobo et al., 2020) [10]. Indeed, e-learning platforms have come to play a key role in this pandemic.

### 1.1. Online Learning: Previous Studies

E-learning programs run on the premise that students and teachers are willing to embrace the change. Their readiness and degree of acceptance also affects the performance of the e-learning program. Across several countries around the world, a great deal of research has discussed problems of e-learning. The online learning process is defined by the use of technical instruments in linking, interacting and accessing materials in any situation in which both the instructor and learner are physically separated (Mulenga & Marbán, 2020) [11]. This has seen an increased popularity when it comes to people who cannot seek formal education because they are far away from institutions of learning or because they are engaged in work or other activities (Kite et al., 2020) [12]. Technology has been promoted in many universities to use online teaching and learning. Digital teaching and learning have recently become the world's only alternative for many colleges, owing to the epidemic of COVID-19 and virtual quarantine (Alyami, 2020; Brooks & Davis, 2020; Kaufmann & Vallade, 2020) [13–15].

Numerous cases of research have explored the perception of lectures in online education (Abed & Shackelford, 2020; Al-Ahdal, 2020; Al Kurdi et al., 2020; Hazaea et al., 2021; Naim & Alahmari, 2020; Opoku et al., 2020) [1,2,16–19]. The findings of these studies show that on-line teaching involves lecturers in the intensive work of preparation and organization of materials and provides constant feedback to students and the evaluation of online tasks (Ceglie & Black, 2020; Lao & Gonzales, 2005) [20,21]. Online education is often seen by teachers to be versatile and relaxed but time-consuming and labor-intensive, as they have to spend a lot of time training and giving constant input to students (San-Martín et al., 2020; Polly et al., 2020) [22,23]. The concerns affecting the experience of teachers in online learning were clustered in terms of teaching presence and engagement, equality, access and fairness, as well as pedagogy and metacognition (Al-Mahmood and McLoughlin, 2004; Nandi et al., 2020) [24,25]. De Gagne and Walters (2009) [26] suggest that the lack of physical presence is one of the key problems affecting the online teaching experience for teachers. Research shows that many factors influence students too in the online learning environment in the analysis of learner experiences in online learning, including technical issues, costs, internet access, encouragement, social interactions and academic and technical skills (Aljaraideh & Al Bataineh, 2019; Muilenburg and Berge, 2005; Pei and Wu, 2019; Rizvi et al., 2019) [27–30]. The use of interactive technology tools also leads to the successful online learning experience of students, because interactive technologies facilitate their engagement with an instructor and improve their sense of the presence of the student (Al-Fraihat et al., 2020; Al Kurdi et al., 2020; Mohammadi, 2015) [31–33].

College education was expected to change drastically in 2020, as most universities turned to e-learning and teaching, and students actively engaged in better online learning environments and relied heavily on themselves for learning (Chirikov et al., 2020; Kizilcec

et al., 2020) [34,35]. In another study, students reported being distracted in online learning (Napier et al., 2020) [36]. E-learning is a challenge for various universities in both developed and developing countries. Concern for the readiness to adopt and use e-learning software is likely in developing countries, as the literature indicates that substantial progressive steps have already been taken in this field. Eltahir (2019) [37] said that the challenge of introducing e-learning programs in developed countries remains a reality due to the digital divide with developing countries. Abdullah and Ward (2016) [38] investigated the factors influencing e-learning acceptance with TAM. Such findings demonstrated auto-efficiency; student e-learning acceptability plays a significant role in social expectations, happiness, anxiety and machine familiarity. Similarly, Alhabeeb and Rowley (2017) [39] found that university staff are critical in promoting effective e-learning in Saudi Arabic universities through their knowledge of learning technology, computer system students and technological infrastructure. This study aims to provide a new perspective to the existing literature on the key challenges and factors influencing the performance of e-learning in a global context; though numerous studies in the field of e-learning have been done, Saudi Arabia, which can serve as a leading example for developing countries, may contribute to the existing body of knowledge during the COVID-19 Pandemic.

In an exhaustive global survey, Marinoni et al. (2020) [40] opine that it is clear that the future of higher education needs rethinking in many ways. Further, they point out that international and multilateral cooperation within the higher education sector and with policymakers, communities and other stakeholders will need to be increased and strengthened to ensure the functioning of the higher education sector.

Student-related adverse outcomes have been reported across four countries in a study by Cifuentes-Faura et al. (2021) [41]. The study showed that the social support of instructors made available to students was inadequate, with Omani students being the most affected. Employment opportunities went southwards in all of the four countries surveyed, with Nigerian students suffering massive job losses. Finally, students from the four countries were required to put in a lot of effort and energy to fulfil the requirements of the program. To quote another study, Faura-Martínez et al. (2021) [42] found, in a study with more than 3000 Spanish university students, that in the new academic dispensation, households had to invest in ICT equipment, and universities offered support to vulnerable students. Students were not prepared for this change, found it difficult to follow the course online, spent more hours per day studying and achieved lower academic performance.

### 1.2. The Significance of the Study

It is not wholly true that e-education came into vogue only in the wake of the COVID-19 pandemic. A few years prior to its outbreak, several universities in KSA started limited e-learning programs. However, the complete shift to the modality during and after the pandemic did bring to the fore many primary concerns and factors affecting the use of e-learning. The lack of an e-learning framework limits the advantages of an e-learning program according to the students' willingness and motivation to use it, unless the opinions of students and teachers are not fully explored. E-learning allows universities to better understand the students' needs, leading to effective e-learning programs. As we know, the conditions brought about by the pandemic made it urgent to discuss thoroughly the problems and factors affecting the use of online training, even though e-learning programs had already been introduced in various universities across Saudi Arabia. The study will be beneficial for curriculum planners, university teachers and students to realize the vision of the Kingdom of Saudi Arabia in 2030, particularly with respect to online education. Practically, the findings of the proposed study will guide policymakers to implement higher funding for public universities giving priory to online education. It is expected that this work will form the educational framework from a social as well as economic perspective.

*1.3. Research Objectives*

This study aims to identify and examine the main challenges and experiences of internal stakeholders, such as the teachers and students with online education during the transition of campus education to e-learning. The key emphasis of this proposed study was on the following research objectives: (1) Explore the perceptions of the teachers and students on e-learning implementation during the COVID-19 Pandemic; (2) Ascertain the factors affecting the proper implementation of e-learning as perceived by the professors and students; (3) Assess the relevance and the skills of the professors and students to engage in e-learning during the pandemic; (4) Propose initiatives to both universities towards e-learning relevance and internationalization.

*1.4. Research Questions*

The research is, therefore, aimed at exploring the key challenges and factors that influenced the use of the e-learning program during the COVID-19 pandemic. It will answer the following research questions:

1.  How do teachers and students describe their background and perception of e-learning?
2.  What factors affect the proper implementation of e-learning as perceived by the teachers and students at the two universities?
3.  How relevant and updated are the skills of the teachers and students to engage in e-learning during special circumstances, such as the pandemic?
4.  What initiatives can be recommended to both universities towards e-learning relevance and internationalization?

**2. Methods**

The settings of this study were two campuses: Qassim University (QU) and Imam Abdulrahman Bin Faisal University (IAU), Saudi Arabia. The rationale behind collecting data from two universities was that results so obtained would reflect a broader canvas of responses and, hence, justify their universality. The participants were 22 teachers, each from the mainstream academic branches of the Department of English Language and Translation at the universities, viz., Foreign Language Learning (N = 3); Pedagogy, learning, teaching (N = 4); Language teaching (N = 3); Literary Studies (N = 4); Applied Linguistics (N = 4) and TESOL (N = 4). A convenience sample (N = 54) was taken in equal numbers form the two study environments, as the researcher had previously taught these students during short-term programs. All student participants were from the Department of English Language and Translation at the two universities and were exposed to e-learning for at least three months at a stretch in the past one year, with median age falling at 21.7 years.

*2.1. Instruments*

2.1.1. Teachers' Questionnaire

The teachers' questionnaire consisted of 18 items loaded onto 12 factors discussed variously in the following section. Face validity of this questionnaire was ensured by seeking the opinion of a panel of three senior academics in higher education who agreed that the instrument measured the constructs that were intended to answer the research questions in the study. All questions were thereafter checked by a psychometrician colleague to rule out the possibility of any questions being confusing, double-barreled or leading. Given the small sample size of the teacher participants, the questionnaire was not pilot tested. Factor loadings were checked for all the items in this questionnaire, and Cronbach's Alpha was calculated to verify how well the factors correlated. Sadly, two questions from the original questionnaire had to be removed to obtain a value upwards of 0.6.

2.1.2. Students' Questionnaire

The students' questionnaire was designed after a thorough review of available literature on perceptions towards e-learning. The administered questionnaire contained 14 items, with these loading onto three factors. Face validity of this instrument was established by

consulting a panel of three university professors engaged in e-teaching throughout the pandemic period. At least four inconsistencies were pointed out by the panel, two items advised to be removed and three to be rephrased. These operations were duly carried out.

After adjusting the items as per the advice of a psychometrician, the questionnaire was tested with a group of 35 students at the two universities. The test sample shared general characteristics with the study sample in terms of age, academic background and familiarity with the researcher. Factor loadings were checked for the items, and with these being >0.70, the researcher felt assured that the instrument was valid and fit to be used. It may be pointed out here that the questions loading onto the same factors were later aggregated (i.e., combined) and compared during the final data analysis phase, as can be seen by the grouping of such items in a table.

*2.2. Research Approach*

The current study used a hybrid approach to research, consisting of two steps, the collection and analysis of quantitative data and the compilation, analysis and evaluation of quantitative results (Creswell et al., 2011) [43]. The data were collected and analyzed in two sequential phases. E-surveys were produced by the researchers in the first phase and distributed online to lecturers as well as students. The survey comprised the following sections: (i) demographic information for participant's gender, age, country/city, university level and major; (ii) the challenges faced by teachers and students in e-learning (the tools used for e-class access, type of Internet access, the timing for e-classes, lengths, e-classes and tools for delivering online classes) and a third section comprising open-ended questions to elicit a detailed individual response. At the end of this phase, both descriptive and inferential data, including the average values and correlations, were analyzed. The second phase of data collection began on the basis of the results of the survey. Throughout this step, the Protocols were structured to gather qualitative data that elaborated and supported quantitative findings for follow-up interviews through WhatsApp and focus group discussions through Google Meet. The participants were requested for their mobile numbers and e-mails (kept confidential) for the collection of qualitative data. The qualitative data was analyzed with thematic and content analyses. This process was imperative, since it took place in a variety of steps: recording audio and video from subsequent interviews and conversations in the focus group, reading and coding transcripts in line with study questions, taking notes and debating agreement between code-makers and classifying codes into patterns or topics.

## 3. Data Analysis and Results

A descriptive analysis was used to analyze the quantitative data. The frequencies and percentages were obtained. For analyzing the qualitative data, a thematic analysis was applied.

Interviews were conducted in the second phase of this study, wherein both students and teachers brought out the reasons why and how technology and infrastructure-related factors adversely affected the implementation of e-learning. These interviews were via the Google Meets app, with prior appointments with five teachers and 11 students. These were purely voluntary and, hence, the small numbers. Sentence was the level of the analysis. Findings were presented by showing the teacher (Teacher No.) and student number (Student No.) along with each theme.

Research Question 1: How do teachers and students describe their background and perception of e-learning?

As to the perception of the teachers of e-learning during the COVID-19 pandemic, based on the results of the data shown in Table 1 and Figure 1, about half (55%) of the teacher-respondents perceived that it was *difficult to teach online*. Looking at the break-up of values, 18% said it was *very difficult to teach online*, while 14% of them said it was *moderately difficult to teach online*. Only 9% of them said it was *not difficult to teach online*, and only 5% of them affirmed that it was *not difficult at all*. The results of the study imply that teachers were

ambivalent on the transition from face-to-face learning to online learning during COVID-19, as an indication of their sentiments expressed that it is difficult to teach online. This may be attributed to the reality that teachers seem hard to keep up with online instruction. In the series interview conducted with them to validate the finding of the study, Teacher 1 said "*we find it difficult to teach through online because our standards are restricted. We are not able to see if our students are really leaning*." Teacher 3 expressed "*At our age, online technology for us is hard to follow particularly in using the online technology learning platform.*" In a like manner, Teacher 5 shared, "*low internet connectivity makes us slow to process learning.*" From the above and other responses duly transcribed from the recorded interviews, it is apparent that teachers faced many technological and infrastructural challenges, such as slow internet, frequent breaks in connection and a lack of backend support. The other challenge they faced to some extent was that of students losing interest in the classes when they find teachers struggling with the tech.

**Table 1.** Responses of Teachers on the e-learning during the Pandemic.

| Scheme 22 | Frequency (N = 22) | Percentage |
|---|---|---|
| Very Difficult to teach online | 4 | 18 |
| Difficult to teach online | 12 | 55 |
| Moderately Difficult to teach online | 3 | 14 |
| Not Difficult to teach online | 2 | 9 |
| Not difficult at all | 1 | 5 |

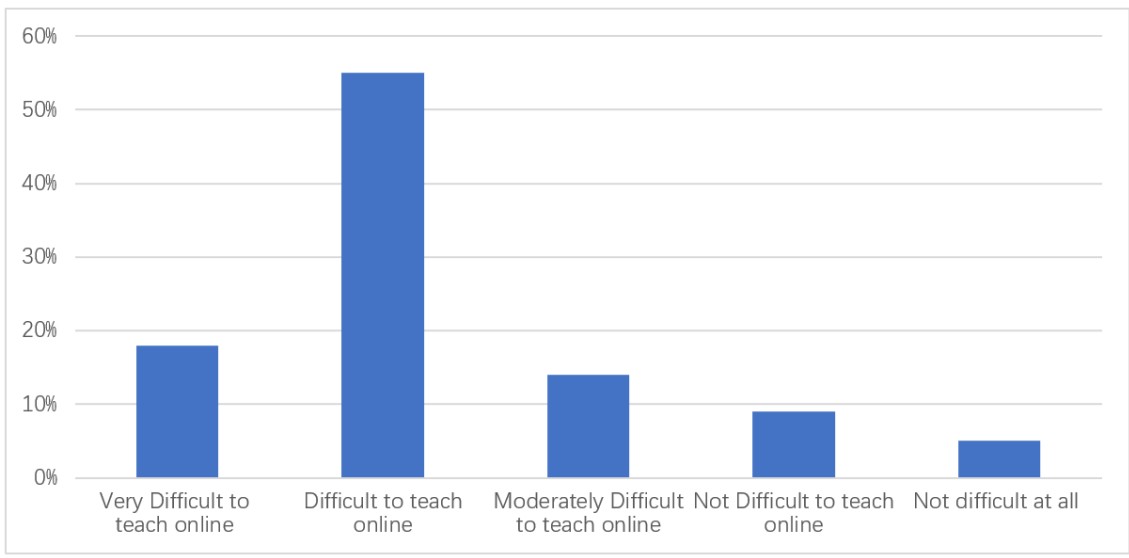

**Figure 1.** Responses of Teachers on the e-learning during the Pandemic.

To sum up, most of them claimed that time management is difficult, since students do not have the same level of privileges. Some of them can have online learning at a fast mode, while others find it difficult to log into the leaning portal.

Consequently, the perception of students of e-learning during the pandemic, presented in Table 2 and Figure 2, showed that the majority, or 43%, of the students face difficulty with online learning. A total of 20% of them reported the difficulty to be moderate, and 15% said it was very difficult for them to learn online. A very small proportion of the respondents, at 13%, reported no difficulty in e-learning, whereas 9% opined that it was not difficult at all. The second phase of data collection, viz., interview, identified the reasons why learners found e-learning 'very difficult'. Students' interviews identified the following themes: (i) Poor internet; (ii) Monotony of lessons; (iii) Inability to connect with the teacher on a one-to-one basis; (iv) Teachers' inability to process students' queries.

**Table 2.** Responses of Students on the e-learning during the Pandemic.

| Statements | Frequency (N = 54) | Percentage |
|---|---|---|
| Very Difficult to learn online | 8 | 15 |
| Difficult to learn online | 23 | 43 |
| Moderately Difficult to learn online | 11 | 20 |
| Not Difficult to learn online | 7 | 13 |
| Not difficult at all | 5 | 9 |

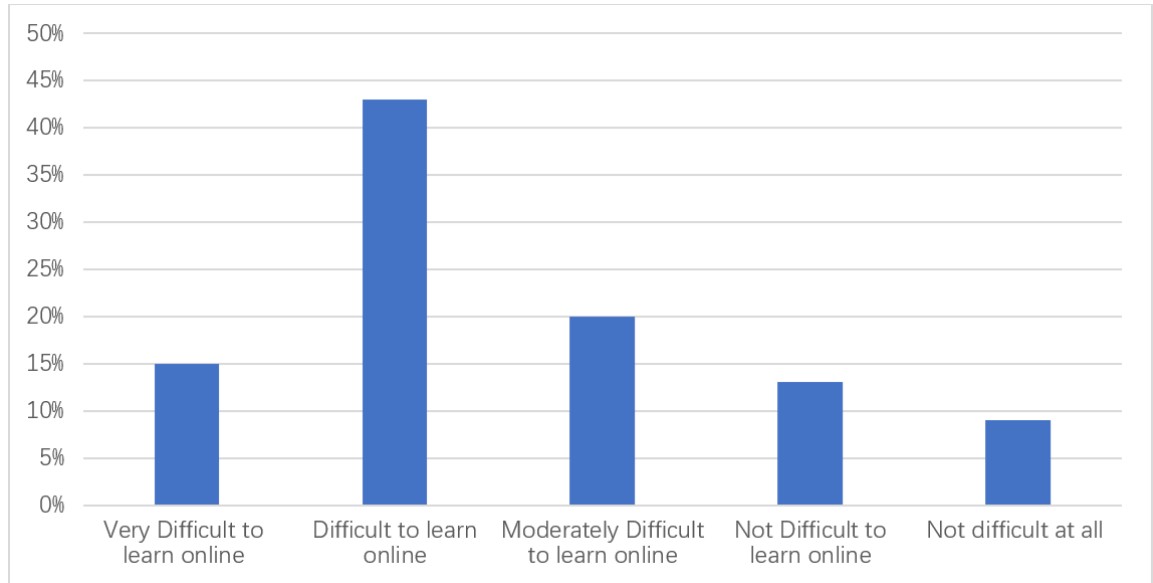

**Figure 2.** Responses of students on the e-learning during the Pandemic.

Students 1 reported, *"We find it difficult due to limited access to internet and other technology tools/"*. Student 10 also shared, *"We struggle online because it's hard to manage our time, we are bombarded with so many requirements that are hard to do."* In addition, Student 4 said, *"Studying at home is solitary. Without the buzz of the classroom and our coworkers, it is no wonder why many of us start experiencing a deep sense of isolation that is steadily eroding the enthusiasm for learning."* However, Student 9 respondent shared: *"At home I find many things hard to learn because of many household chores I need to attend to. I also take care of my siblings when my parents go to work."* Finally, to sum up, most of the students found e-learning difficult because the workload in online learning is higher than the regular face-to-face class. Similarly, they need to finish their requirements within the deadline. Moreover, the majority of the students voiced opinions affirming their *capability to use the online learning platform, but they do not have the enough technology tools to learn. I, for one, as using a cellphone, and it's difficult to upload requirements, unlike using laptops where students can easily make their requirements."*

Four of the interviewed students pointed out that the e-learning platforms were plagued with distractions, ranging from technological to domestic disturbances. Two students pointed out that the content was flat and unengaging, as they were only expected to listen to the lectures with their cameras and microphones turned off to ensure the least burden on the internet connection. This encouraged students to wander off from the lessons, and the teachers were forced to frequently interrupt their teaching to check this trend. Moreover, they pointed out that students engaged in peer messaging, leading to unwarranted distraction in the learning process. In other words, overall, the e-learning experience failed to engage them, as the teacher had no means of watching the students, and the motivation to learn was at a low due to the pandemic and students' lack of autonomy.

One student suggested that class sizes needed to be reduced if e-learning was to be ensured in the Saudi context, as the students here are not autonomous, and they need constant motivation from the instructors to stay connected in the learning process.

However, another student noted that the availability of ready-to-use notes, which the teacher was mandated to upload in the e-learning process, took away the sense of responsibility in the students to work for their success, and they tended to avoid classes, thinking that the teacher could not see or hear them and that the notes would be sufficient to see them through the exams.

Research Question 2: What factors affect the proper implementation of e-learning as perceived by the teachers and students at the two universities?

As to the factors affecting the proper implementation of e-learning as perceived by both students and teachers, Table 3 and Figure 3 show that technology- and infrastructure-related problems were the biggest delineating factors, as reported by 50% of the teacher-respondents. Table 4 and Figure 3 also show that in the students' perception, similar problems were faced, where technology- and infrastructure-related obstacles were reported as being limiting by 52% of the student-respondents. The finding shows congruence on the responses of both students and teachers, where technology-related problems are a factor to manage in the Saudi e-learning system.

**Table 3.** Responses of Teachers on the factors affecting the proper implementation of e-learning during the Pandemic.

| Statements | Frequency (N = 22) | Percentage |
|:---:|:---:|:---:|
| Teacher-related factors | 7 | 32 |
| Student-related factors | 4 | 18 |
| Technology- and Infrastructure-Related Factors | 11 | 50 |

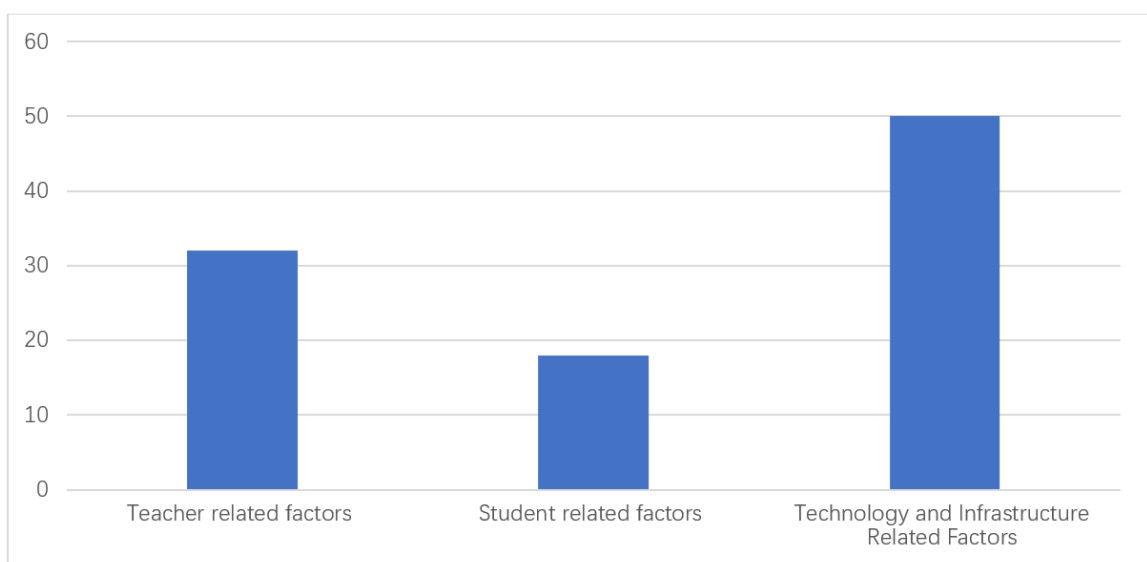

**Figure 3.** Responses of Teachers on factors affecting the proper implementation of e-learning during the Pandemic.

**Table 4.** Responses of Students on the factors affecting the proper implementation of e-learning during the Pandemic.

| Statements | Frequency (N = 54) | Percentage |
|:---:|:---:|:---:|
| Teacher-related factors | 15 | 28 |
| Student-related factors | 11 | 20 |
| Technology- and Infrastructure-Related Factors | 28 | 52 |

It can be gleaned from the data in Table 3 and Figure 3 that there were 32% of respondents who reported that teacher-related factors influenced the implementation of

e-learning during the pandemic, while a relatively low 18% went for student-related factors. Similarly, with the students' responses as depicted in Table 4 and Figure 4, 28% of the student-respondents said that it was attributable to teacher-related factors, and only 20% said it was caused by student-related factors.

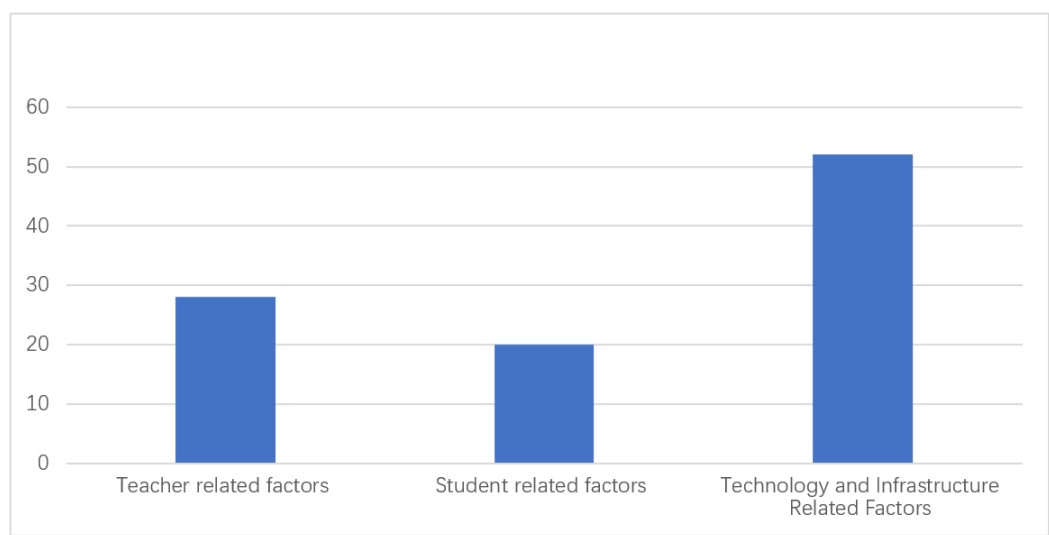

**Figure 4.** Responses of Students on the factors affecting the proper implementation of e-learning during the Pandemic.

Research Question 3: How relevant and updated are the skills of the teachers and students to engage in e-learning during special circumstances, such as the pandemic?

As to the relevance of the skills of both teachers and students in online learning during special circumstances, such as the COVID-19 pandemic, Table 5 and Figure 5 show that a majority of the teachers opined that their skills were only moderately relevant and moderately updated, as shown from the 50% responses; that as many as 23% of them reported that they were relevant and updated; only 14% of the respondents believed that their skills were both very relevant and very updated and an equal number (14%) reported they were irrelevant and not updated. This result shows that teachers had not yet attained the level of technological competence expected of them, which needs serious attention from the government and curriculum planners, who may do well to take steps for in-service skill-upgradation.

**Table 5.** Responses of Teachers.

| Statements | Frequency (N = 54) | Percentage |
| --- | --- | --- |
| Very relevant and very updated | 3 | 14 |
| Relevant and updated | 5 | 23 |
| Moderately relevant and moderately updated | 11 | 50 |
| Irrelevant and not updated | 3 | 14 |
| Very irrelevant and not at all updated | 0 | 0 |

As to the responses of the students shown in Table 6 and Figure 6, most of the student-respondents admitted that their skills were relevant and updated in the using e-learning, at 50%. There were 22% of them who reported their skills for e-learning to be moderately relevant and moderately updated, 20% reported them to be very relevant and very updated and 7% of them said they were irrelevant and not updated vis-à-vis e-learning. Furthermore, it is also good to note that none of them perceived their skills to be very irrelevant and not at all updated, which shows that all of them possess at least some of the relevant skill set. As shown from the findings, the students manifested a higher level of competence in

using the e-learning technologies than their teachers, which is an indication that students nowadays are truly digital natives.

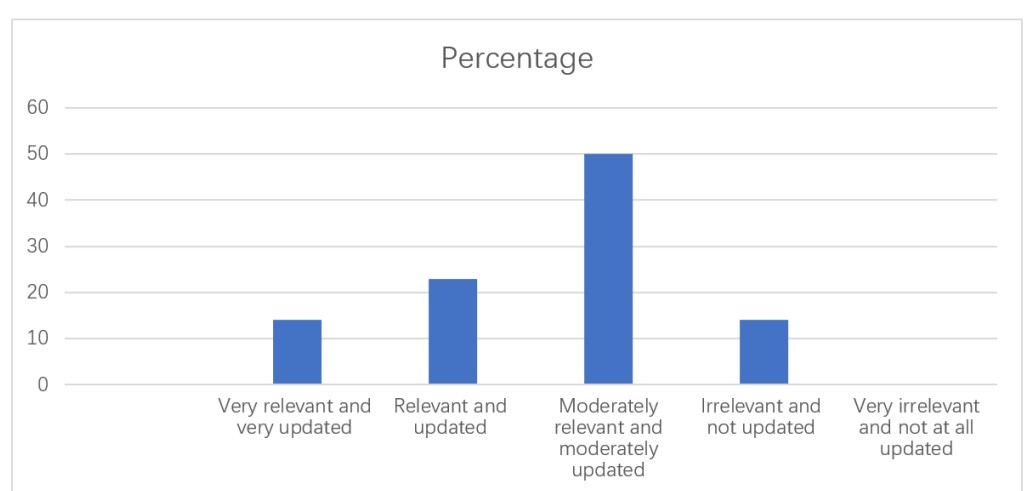

**Figure 5.** Responses of Teachers.

**Table 6.** Responses of Students.

| Statements | Frequency (N = 22) | Percentage |
|---|---|---|
| Very relevant and very updated | 11 | 20 |
| Relevant and updated | 27 | 50 |
| Moderately relevant and moderately updated | 12 | 22 |
| Irrelevant and not updated | 4 | 7 |
| Very irrelevant and not at all updated | 0 | 0 |

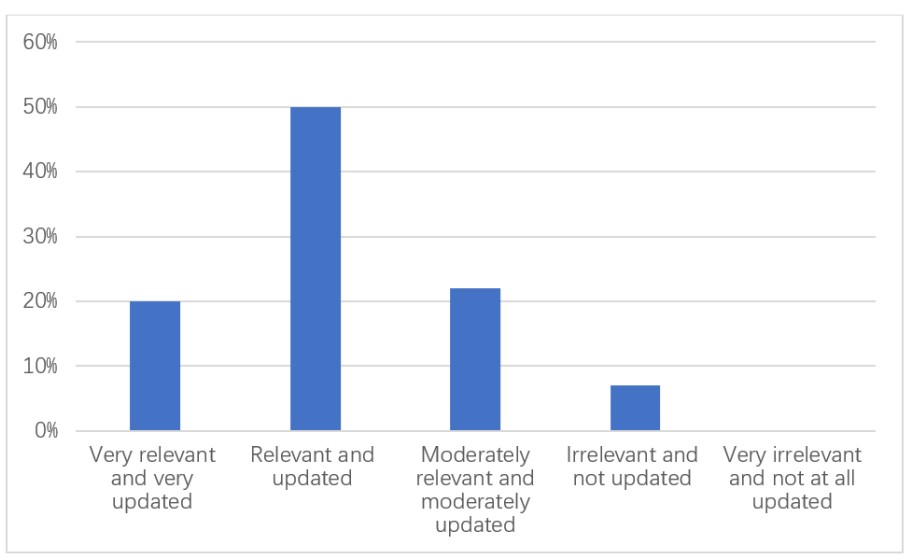

**Figure 6.** Responses of Students.

As to the interviews conducted for both teachers and students, the teachers reasoned out the following: *Teacher 2 reported that "We don't look at ourselves as digitally literate like our students. Sometimes we have the feeling of discomfort to cope with the technology tools to leaning."* Teacher 4 shared, *"I felt the uneasiness in using a computer; I have the fear that I might destroy it when using."* Teacher 3 opined, *"I felt I am embarrassed when my students are better than me in using computers. I am afraid that my students may judge me as computer illiterate."* However, Teacher 5 responded, *"We have little knowledge with communication technology. We also have*

*difficulties memorizing protocols when utilizing applications and operating devices. In addition, there are teachers that claim they can't adapt to computer technology."*

Research Question 4: What initiatives can be recommended to both universities towards e-learning relevance and internationalization?

This study also ascertained the recommendations of the participants on the possible initiatives of online learning towards the internationalization of Saudi universities. Both students and teachers strongly recommended all the items presented in Table 7. As to the order of priorities, the highest priority was reported to be the provision of necessary online technology tools for both teachers and students. This was followed by the demand for relevant online international degrees. Close behind this came the need to improve teachers' and students' technological literacy. The last in this listing was the inclusion of security features on the website for online learning. While these were ranked according to the mean values, all were interpreted to be strongly recommended for the Saudi universities to initiate the ushering in of relevance and internationalization.

**Table 7.** Responses of Teachers.

| Statements | Mean | Interpretation | Rank |
|---|---|---|---|
| Provide necessary online technology tools for both teachers and Students | 4.40 | Strongly Recommended | 1 |
| Improve teachers' technological literacy | 4.27 | Strongly Recommended | 3 |
| Improve students' technological literacy | 4.23 | Strongly Recommended | 5 |
| Offer online relevant international degrees | 4.30 | Strongly Recommended | 2 |
| Development of user-friendly learning platforms | 4.26 | Strongly Recommended | 4 |
| Include a security feature of the website for online learning | 4.22 | Strongly Recommended | 6 |

**Legend:** Strongly Recommended (4.20–5.00); Recommended (3.40–4.19); Undecided (2.60–3.39); Not Recommended (1.80–2.59); Strongly Not Recommended (1.00–1.79).

*Discussion*

This aim of this study was to examine and identify the main challenges and experiences of internal stakeholders, such as the teachers and students, on online education during the transition to e-learning made imminent by COVID-19 on all Saudi campuses. Findings of the study showed that students and teachers perceived online learning as difficult. They faced serious problems in dealing with the online learning mode, because this learning platform is a new learning transition. This finding has been confirmed by many previous studies (e.g., Al-Ahdal, 2020; Algahtani et al., 2020; Hazaea et al., 2021; Muhammad et al., 2020; Naim and Alahmari, 2020) [1–4,18]. They reported that online learning challenged many universities during the COVID-19 pandemic era. Furthermore, they affirmed that the abrupt transition to e-learning raised difficulties for university teachers and students. Students were not able to take to e-classes readily.

For the second query of the study, it was evident that there are factors affecting the proper implementation of online learning. The biggest problem which prevented the implementation of e-learning were the technology- and infrastructure-related problems, which were the biggest delineating factors, followed by students and teachers' factors. This finding can be interpreted as indicating that the availability of technical devices and the strong infrastructure for applying e-learning in the Saudi Arabia are still insufficient. Further, similar findings have been reported by researchers, indicating that teachers are faced with problems and dilemmas in dealing with online learning due to student-related, teacher-related and technology-related factors. According to Gyampoh et al. (2020) [44], the mindset, technical knowledge, time limitations, pedagogy and professional concerns were all core elements of distance learning. In addition, there were also problems with the levels of web-based training readiness and satisfaction among teachers. Most answers were considered strong in terms of preparation, temperature, and ability to lecture online on course design, connectivity, time control and technical issues (Kaden, 2020; Van Nuland et al., 2020; Wu et al., 2020) [45–47].

Cifuentes-Faura et al. (2021) [41] showed that the social support of instructors made available to students was inadequate, with Omani students being the most affected. Moreover, Faura-Martínez et al. (2021) [42] found that it is difficult to follow the course online and spend more hours per day studying and that they achieved lower academic performance.

The third question of this study interrogated the readiness of students and teachers' skills to engage in e-learning during the pandemic. The study found that half of the teachers reported their moderately relevant and updated skills, whereas the majority of students showed their relevant and updated skills to use online learning. This finding showed that students are more up-to-date with the use of technology than their teachers. Teachers' lack of innate technological capabilities for online teaching has been reported as a challenge for institutions of higher education in a study by García-Morales et al. (2021). Alhabeeb and Rowley (2017) [39,48] found that university staff are critical in promoting effective e-learning in Saudi Arabic universities through their knowledge of learning technology, computer system students and technological infrastructure. Furthermore, previous surveys have also shown that the faculty has adverse opinions on the use of software on the internet and also a pessimistic outlook on online learning (Al-Ahdal, 2020; Borup et al., 2019; Erdemir and Ekşi, 2019; Parsons et al., 2019; Zitouni et al., 2021) [2,49–52]. Moreover, most teachers declined to consider the idea of reforming the traditional teaching method by switching to e-learning. Other studies covered barriers to online training in the areas of an operational, training, technology and cost–benefit study (Martin et al., 2020; Nimasari et al., 2019; Salayo et al., 2020; Zheng et al., 2020) [53–56]. The findings of this study suggest that problems and challenges to online education must be resolved and online courses carefully planned and managed. In order to improve instructor attitudes about online teaching and thereby reduce their resistance to online schooling, the unwelcome ideas about online teaching generally kept by teachers must be updated or corrected. In addition, the teachers' sense of commitment to the online courses has been shown to be associated with both the social experience and the teaching experience (Hasan and Bao, 2020) [57]. Some complexities function well in face-to-face but just not in online learning. The faculty should be told how to promote student engagement and ensure success in a virtual environment (Kim et al., 2020) [58]. The study sheds light on the challenges of students and teachers in online learning, mainly to do with the presence of technology tools to be used during online learning (Al-Saggaf and Rosli, 2021; Antonaci et al., 2019′ Dhawan, 2020; Gunasinghe et al., 2019) [59–62]. This means that administrations need to provide facilitating conditions to allow students and teachers to avail the maximum benefits of technology-enabled learning. Facilitation requirements ensure that the infrastructure that enables the use of the suggested system is comfortable and accessible. The ease of usage would then help assess the availability of technical and infrastructural resources, allowing students and teachers to successfully adopt the eLearning process. The perceived simplicity of usage and the perceived effectiveness have a big influence on the useful practices, which is a motivating influence (Bervell and Arkorful, 2020; Hasani et al., 2020; Shen et al., 2019) [63–65].

Finally, the study explored initiatives that can be recommended to both universities towards e-learning relevance and internationalization by providing the necessary online technology tools for both teachers and students; improving students and teachers' technological literacy; offering online relevant international degrees and developing user-friendly learning platforms, including security features of websites for online learning. With the recommendations presented by teachers and students, it is expected that the Saudi universities will be able to attain a global reputation, as they will transition to the new normal of education. It is apparent that very few colleges across the globe claim they are global universities where their students and staff from numerous nationalities also belong to them (Erkkilä and Piironen, 2020) [66]. Instead, several universities can be listed as international, multinational or national. The provision of an effective e-learning platform is a way to promote the quality of universities towards internationalization.

## 4. Conclusions

This research attempts to examine and recognize the principal problems and perspectives of online teachers and students during the transformation of universities to COVID-19. Results of the study showed that both teachers and students face difficulty in e-learning due to technology-related, teacher-related and student-related factors. In the current study, the teachers rated themselves moderately competent on the use of technology tools for online learning, while students assessed themselves as competent. These issues plague many educational settings, including KSA, even though initiatives were taken here by both students and teachers to improve the transition of the universities to online learning, as well as to promote the quality of universities towards internationalization, particularly during this COVID-19 Pandemic, by providing the necessary online technology tools for both teachers and students, improving students and teachers' technological literacy, offering online relevant international degrees, developing of user-friendly learning platforms and activating the security features of the websites used in the online learning.

*Recommendations, Implications and Future Research Direction*

Based on the conclusion of the present study, the following are recommended: (1) Saudi universities should seriously take action on the needs of teachers and students towards effective implementation of e-learning; (2) the acquisition of state-of-the-art technology tools for both students and teachers will help them to become global universities; (3) the conduct of professional development growth for teachers, focusing on their attitude and competence in e-learning modalities, should be conducted; (4) online counseling programs of the universities should be conducted to lessen the boredom the students felt during homeschooling. The study was conducted with a small number of faculty and student samples. Hence, a follow-up study of a larger scope should be conducted to validate the present study findings. It is also suggested that larger samples be included, as well as other important variables, so as to have a more conclusive finding. The use of other research methods is encouraged. The study has implications for curriculum implementers and designers towards a revolutionized education.

**Author Contributions:** Conceptualization, M.S.; methodology, A.A.; validation, F.I.A.A.; formal and analysis A.A.; investigation, K.L.; resources, M.S.; data curation, M.S.; writing—original draft preparation, A.A.; writing—review and editing, A.A. & M.S.; visualization, K.L.; supervision, M.S. All authors have read and agreed to the published version of the manuscript.

**Funding:** The authors gratefully acknowledge Qassim University represented by the Deanship of Scientific Research, on the financial support for this research under the number mcs-ths-2022-1-3-I-10362 during the academic year 1441AH/2020AD.

**Institutional Review Board Statement:** The study was conducted and approved by Ethics Committee of Department of English and Translation, Methnab, Qassim University (SEC 2, 2021; 16 February 2021).

**Informed Consent Statement:** Informed consent was obtained from all subjects involved in the study.

**Conflicts of Interest:** The authors declare no conflict of interest.

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
