# Peer review of "Teachers and Learners’ Perceptions of E-Learning Implementation in Special Times: Evaluating Relevance and Internationalization Prospects at Saudi Universities"

_sustainability, doi:10.3390/su14106063_

Round 1
Reviewer 1 Report
The article deals with the interesting subject of perceptions of e-learning implementation during COVID-19. Unfortunately, the methodology contains serious flaws. Some major issues are listed below:
-the methodology section should be a separate section, not just a subsection of the introduction. The title of the subsection Research methodology contains the “:” in the title, which is not appropriate. The research methodology subsection is closer to the research context than the methodology in the current state.
- The crucial methodology information is missing. The methodology part should cover the research questions, participants, instruments, research context, data analysis, results, and other parts related to conducting the research. The participants' background and the instruments analysis are completely missing.
-In the analysis of the research question one, the authors claim, “a large percentage (55%) of the teacher-respondents perceived…”. Since the n=22, the statement “large percentage” is not appropriate. Analysis of students’ answers has the same problem. Data analysis is based on frequencies only.
-Interview analysis is poorly described and is not following any formal procedure related to interviews. For example, p7, l.244, in only 8 lines of text is dedicated to interviews’ analysis.
-The results are interpreted based on the frequencies of the answers to the survey with the low number of participants and poorly described interviews. Practically, there is no statistical analysis. The conclusions cannot be supported based on such analysis.
I suggest the authors improve the paper based on the above-mentioned suggestions.
Author Response
All changes have been done and highlighted as per instructions

Reviewer 2 Report
The paper is interesting and deals with an important issue. However, there are many shortcomings.
The authors should include more literature on Covid-19 and the impact on teachers and students. For example:
Faura-Martínez, U., Lafuente-Lechuga, M., & Cifuentes-Faura, J. (2021). Sustainability of the Spanish university system during the pandemic caused by COVID-19. Educational Review, 1-19.
Marinoni, G., Van’t Land, H., & Jensen, T. (2020). The impact of Covid-19 on higher education around the world. IAU global survey report, 23.
Cifuentes-Faura, J., Obor, D. O., To, L., & Al-Naabi, I. (2021). Cross-cultural impacts of COVID-19 on higher education learning and teaching practices in Spain, Oman, Nigeria and Cambodia: A cross-cultural study. Journal of University Teaching & Learning Practice, 18(5), 8.
García-Morales, V. J., Garrido-Moreno, A., & Martín-Rojas, R. (2021). The transformation of higher education after the COVID disruption: Emerging challenges in an online learning scenario. Frontiers in Psychology, 12, 196.
The conclusions and discussions should be expanded, and the results should be compared with previous work in the literature. It should indicate what is new about this work and how it differs from other published work. My biggest problem is with the methodology. I think it is very basic and does not contribute anything important. Also, the sample is very small and not very representative. There are only 54 students and 22 teachers. What percentage does this represent of the total? An ANOVA or a more extensive methodology should be carried out. This is very descriptive and basic.
Author Response
All changes have been done and highlighted as per instructions.

Reviewer 3 Report
The article discusses the challenges of online learning during the pandemic through a series of surveys and interviews with the various stakeholders. In general, the study is complete and would be useful for readers who are interested in online learning under such a context. Two minor comments:
1) Why is Figure 5 a 3D plot (when the rest of the figures are 2D plots)?
2) I also would like to refer the authors to this article: https://arxiv.org/abs/2009.02705 which similarly discusses the challenges of implementing a online learning environment during the pandemic. I believe the points raised could help to strengthen the arguments made by the authors in their own article.
Author Response

(The authors gave the same response as above.)

Round 2
Reviewer 1 Report
The authors have improved the manuscript according to the comments.
Reviewer 2 Report
The authors have improved the manuscript according to the comments